# Deep RGB-D Canonical Correlation Analysis For Sparse Depth Completion

**Yiqi Zhong**[*]
University of Southern California
Los Angeles, California
yiqizhon@usc.edu

**Cho-Ying Wu**[*]
University of Southern California
Los Angeles, California
choyingw@usc.edu

**Suya You**
US Army Research Laboratory
Playa Vista, California
suya.you.civ@mail.mil

**Ulrich Neumann**
University of Southern California
Los Angeles, California
uneumann@usc.edu

## Abstract

In this paper, we propose our Correlation For Completion Network (CFCNet), an end-to-end deep learning model that uses the correlation between two data sources to perform sparse depth completion. CFCNet learns to capture, to the largest extent, the semantically correlated features between RGB and depth information. Through pairs of image pixels and the visible measurements in a sparse depth map, CFCNet facilitates feature-level mutual transformation of different data sources. Such a transformation enables CFCNet to predict features and reconstruct data of missing depth measurements according to their corresponding, transformed RGB features. We extend canonical correlation analysis to a 2D domain and formulate it as one of our training objectives (i.e. 2d deep canonical correlation, or "2D$^2$CCA loss"). Extensive experiments validate the ability and flexibility of our CFCNet compared to the state-of-the-art methods on both indoor and outdoor scenes with different real-life sparse patterns. Codes are available at: https://github.com/choyingw/CFCNet.

## 1 Introduction

Depth measurements are widely used in computer vision applications [1, 2, 3]. However, most of the existing techniques for depth capture produce depth maps with incomplete data. For example, structured-light cameras cannot capture depth measurements where surfaces are too shiny; Visual Simultaneous Localization And Mappings (VSLAMs) are not able to recover depth of non-textured objects; LiDARs produce semi-dense depth map due to the limited scanlines and scanning frequency. Recently, researchers have introduced the sparse depth completion task, aiming to fill missing depth measurements using deep learning based methods [4, 5, 6, 7, 8, 9, 10]. Those studies produce dense depth maps by fusing features of sparse depth measurements and corresponding RGB images. However, they usually treat feature extraction of these two types of information as independent processes, which in reality turns the task they work on into "multi-modality depth prediction" rather than "depth completion." While the multi-modality depth prediction may produce dense outputs, they fail to fully utilize observable data. The depth completion task is unique in that part of its output is already observable in the input. Revealing the relationship between data pairs (i.e. between observable depth measurements and the corresponding image pixels) may help complete depth maps by emphasizing the information from image domain at the locations where the depth values are non-observable.

---

[*]Both authors contributed equally to this work.

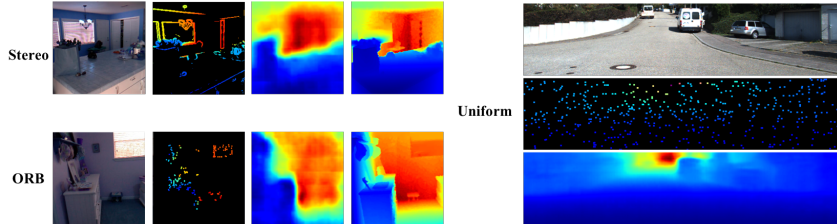

Figure 1: Sample results of CFCNet on different sparse patterns. We show in the order of RGB images, sparse depth maps with different sparse patterns, and dense depth maps completed by CFCNet. For the stereo pattern and the ORB pattern, we show the depth groundtruth in the last column.

To accomplish the depth completion task from a novel perspective, we propose an end-to-end deep learning based framework, Correlation For Completion Network (CFCNet). We view a completed dense depth map as composed of two parts. One is the sparse depth which is observable and used as the input, another is non-observable and recovered by the task. Also, the corresponding full RGB image of the depth map can be decomposed into two parts, one is called the sparse RGB, which holds the corresponding RGB values at the observable locations in the sparse depth. The other part is complementary RGB, which is the subtraction of the sparse RGB from the full RGB images. See Figure 2 for examples. During the training phase, CFCNet learns the relationship between sparse depth and sparse RGB and uses the learned knowledge to recover non-observable depth from complementary RGB.

To learn the relationship between two modalities, we propose a 2D deep canonical correlation analysis ($2D^2CCA$). In the proposed method, our $2D^2CCA$ tries to learn non-linear projections where the projected features from RGB and depth domain are maximally correlated. Using $2D^2CCA$ as an objective function, we could capture the semantically correlated features from the RGB and depth domain. In this fashion, we utilize the relationship of observable depth and its corresponding non-observable locations of the RGB input. We then use the joint information learned from the input data pairs to output a dense depth map. The pipeline of our CFCNet is shown in Figure 2. Details of our method are described in Section 3. The main contributions of CFCNet can be summarized as follows.

- Constructing a framework for the sparse depth completion task which leverages the relationship between sparse depth and its corresponding RGB image, using the complementary RGB information to complement the missing sparse depth information.
- Proposing the $2D^2CCA$ which forces feature encoders to extract the most similar semantics from multiple modalities. Our CFCNet is the first to apply the two-dimensional approach in CCA with deep learning studies. It overcomes the small sample size problem in other CCA based deep learning frameworks on modern computer vision tasks.
- Achieving state-of-the-art of the depth completion on several datasets with a variety of sparse patterns that serve real-world settings.

## 2 Related Work

**Sparse Depth Completion** is a task that targets at dense depth completion from sparse depth measurements and a corresponding RGB image. The nature of sparse depth measurements varies across scenarios and sensors. Sparse depth generated by the stereo method contains more information on object contours and less information on non-textured areas [11]. LiDAR sensors produce structured sparsity due to the scanning behavior [12]. Feature based SLAM systems (such as ORB SLAM [13]) only capture depth information at the positions of corresponding feature points. Besides these most popular three patterns, some other patterns have also been studied. For instance, [14] uses a line pattern to simulate partial observations from laser systems; [15] culls the depth data of shiny surfaces area out of the dense depth map to mimic commodity depth cameras' output. [8] uses uniform grid patterns. The latter appears a simplified and artificial pattern. Real-life situations require a more practical tool.

As for input sparsity, [4] stacks sparse depth maps and corresponding RGB images together to build a four-channel (RGB-D) input before fed into a ResNet based depth estimation network. This treatment produces better results than monocular depth estimation with only RGB images. Other studies

involve a two-branch encoder-decoder based framework which is similar to those used in RGB-D segmentation tasks [9, 10, 16, 17]. Their approaches do not apply special treatments to the sparse depth branch. They work well on the dataset where sparsity is not extremely severe, e.g. KITTI depth completion benchmark [6]. In most of the two-branch frameworks, features from different sources are extracted independently and fused through direct concatenations or additions, or using features from RGB branch to provide an extra guidance to refine depth prediction results.

**Canonical Correlation Analysis** is a standard statistical technique for learning the shared subspace across several original data spaces. For two modalities, from the shared subspace, each representation is the most predictive to the other representation and the most predictable by the other [18, 19]. To overcome the constraints of traditional CCA where the projections must be linear, deep canonical correlation analysis (DCCA) [20, 21] has been proposed. DCCA uses deep neural network to learn more complex non-linear projections between multiple modalities. CCA, DCCA, and other variants have been widely used on multi-modal representation learning problems [22, 23, 24, 25, 26, 27, 28].

The one-dimensional CCA method suffers from the singularity problem of covariance matrices in the case of high-dimensional space with small sample size (SSS). Existing works have extended CCA to a two-dimensional way to avoid the SSS problem. [29, 30, 31] use a similar approach to building full-rank covariance matrices inspired by 2DPCA [32] and 2DLDA [33] on the face recognition task. However, those studies do not approximate complex non-linear projections as [20, 21] attempt. Our CFCNet is the first to integrate two-dimensional CCA into deep learning frameworks to overcome the intrinsic problem of applying DCCA to modern computer vision tasks, detailed in Section 3.2.

# 3 Our Approach

Our goal is to leverage the relationship of the sparse depth and their corresponding pixels in RGB images in order to optimize the performance of the depth completion task. We try to complement the missing depth components using cues from RGB domain. Since CCA could learn the shared subspace with its predictive characteristics, we estimate the missing depth component using features from RGB domain through CCA. However, traditional CCA has SSS problem in modern computer vision task, detailed in Section 3.2. We further propose the $2D^2CCA$ to capture similar semantics from both RGB/depth encoders. After encoders learning the semantically similar features, we use a transformer network to transform features from RGB to depth domain. This design not only enables the reconstruction of missing depth features from complementary RGB information but also ensures semantics similarity and the same numerical range of the two data sources. Based on this structure, the decoder in CFCNet is capable of using the reconstructed depth features along with the observable depth features to recover the dense depth map.

## 3.1 Network Architecture

Proposed CFCNet structure is in Figure 2. CFCNet takes in sparse depth map, sparse RGB, and complementary RGB. We use our Sparsity-aware Attentional Convolutions (SAConv, as shown in Figure 3) in VGG16-like encoders. SAConv is inspired by local attention mask [34]. Harley *et al.* [34] introduces the segmentation-aware mask to let convolution operators "focus" on the signals consistent with the segmentation mask. In order to propagate information from reliable sources, we use sparsity masks to make convolution operations attend on the signals from reliable locations. Difference of our SAConv and the local attention mask is that SAConv does not apply mask normalization. We avoid mask normalization because it affect the stability of our later $2D^2CCA$ calculations due to the numerically small extracted features it produces after several times normalization. Also, similar to [6], we use maxpooling operation on masks after every SAConv to keep track of the visibility. If there is at least one nonzero value visible to a convolutional kernel, the maxpooling would evaluate the value at the position to 1.

Most multi-modal deep learning approaches simply concatenate or elementwisely add bottleneck features. However, when the extracted semantics and range of feature value differs among elements, direct concatenation and addition on multi-modal data source would not always yield better performance than single-modal data source, as seen in [35, 17]. To avoid this problem. We use encoders to extract higher-level semantics from two branches. We propose $2D^2CCA$, detailed in 3.2, to ensure the extracted features from two branches are maximally correlated. The intuition is that we want to capture the same semantics from the RGB and depth domains. Next, we use a transformer network

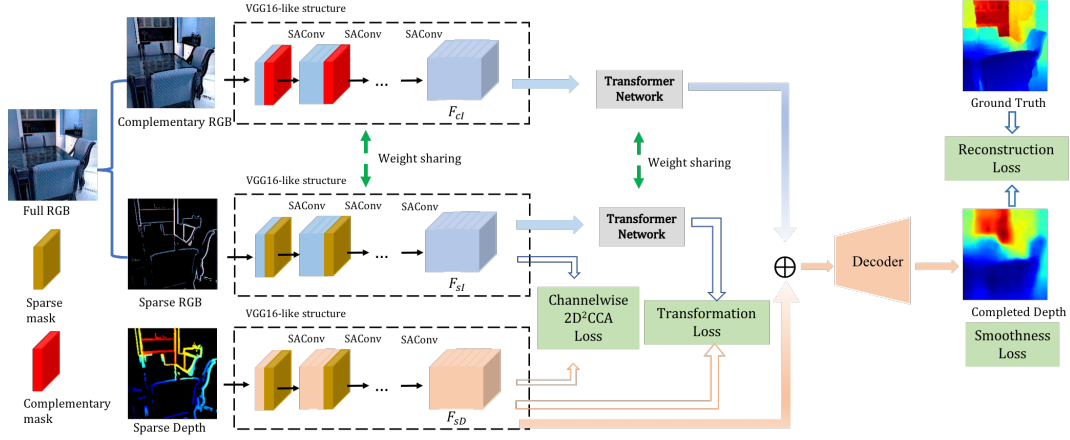

Figure 2: Our network architecture. Here $\oplus$ is for concatenation operation. The input 0 - 1 sparse mask represents the sparse pattern of depth measurements. The complementary mask is complementary to the sparse mask. We separate a full RGB image into a sparse RGB and a complementary RGB by the mask and feed them with masks into networks.

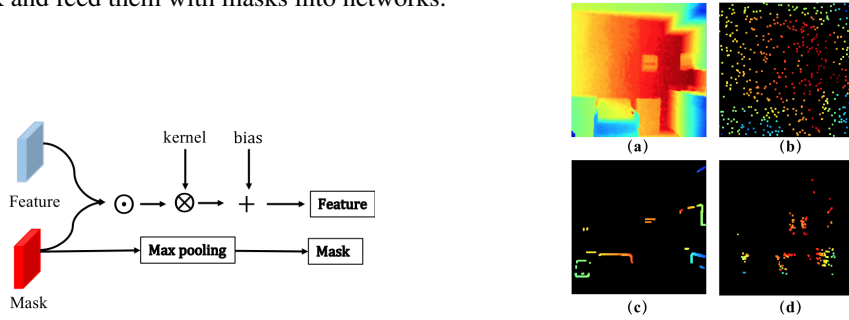

Figure 3: Our SAConv. The $\odot$ is for Hadamard product. The $\otimes$ is for convolution. The $+$ is for elementwise addition. The kernel size is $3 \times 3$ and stride is 1 for both convolution and maxpooling.

Figure 4: Samples of sparsified depth maps for experiments with different sparsifiers. (a) The source dense depth map from NYUv2. (b) With uniform sparsifier (500 points). (c) With stereo sparsifier (500 points). (d) With ORB sparsifier.

to transform extracted features from RGB domain to depth domain, making extracted features from different sources share the same numerical range. During the training phase, we use features of sparse depth and corresponding sparse RGB image to calculate the $2D^2CCA$ loss and transformer loss.

We use a symmetric decoder structure to decode the embedded features. For the input, we concatenate the sparse depth features with the reconstructed missing depth features. The reconstructed missing depth features are extracted from complementary RGB image through the RGB encoder and the transformer. To ensure single-stage training, we adopt weight-sharing strategies as shown in Figure 2.

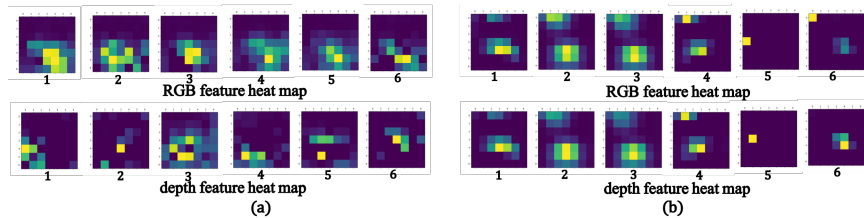

Figure 5: The visualizations of $F_I$ and $F_D$ using an example from NYUv2 dataset. Visuals are heat maps of the extracted features. Brighter color means a larger feature value. The values within a single map were normalized to [0, 1]. (a) The feed-forward heat maps at the first iteration. (b) The feed-forward heat maps after being trained with 10000 iterations. The figure shows the heat maps of the first 6 channels. The numbers under the heat maps represent the channel numbers. The example demonstrates that the $2D^2CCA$ is able to capture similar semantics from different sources.

## 3.2 2D Deep Canonical Correlation Analysis ($2D^2CCA$)

Existing CCA based techniques introduced in Section 2 have limitations in modern computer vision tasks. Since modern computer vision studies usually use very deep networks to extract information from images of relatively large resolution, the batch size is limited by GPU-memory use. Meanwhile, the latent feature representations in networks are high-dimensional, since the batch size is limited, using DCCA with one-dimensional vector representation would lead to SSS problem. Therefore, We propose a novel 2D deep canonical correlation analysis($2D^2CCA$) to overcome the limitations.

We denote the completed depth map as $D$ with its corresponding RGB image as $I$. Sparse depth map in the input and the corresponding sparse RGB image are denoted as $sD$ and $sI$. RGB/Depth encoders are denoted as $f_I$ and $f_D$ where the parameters of the encoders are denoted as $\theta_I$ and $\theta_D$ respectively. As described in Section 3.1, $f_I$ and $f_D$ use the SAConv to propagate information from reliable points to extract features from sparse inputs. We generate 3D feature grids embedding pair ( $F_{sD} \in \mathbb{R}^{m \times n \times C}$, $F_{sI} \in \mathbb{R}^{m \times n \times C}$) for each sparse depth map/image pair $(sD, sI)$ by defining $F_{sD} = f_D(sD; \theta_D)$ and $F_{sI} = f_I(sI; \theta_I)$. Inside each feature grid pair, there are $C$ feature map pairs $\left(F_{sD}^i \in \mathbb{R}^{m \times n}, F_{sI}^i \in \mathbb{R}^{m \times n}\right), \forall i < C$, and $C = 512$ in our network. Rather than analyzing the global correlation between any possible pairs of $(F_{sD}^i, F_{sI}^j), \forall i \neq j$, we analyze the channelwise canonical correlation between the same channel number $\left(F_{sD}^i, F_{sI}^i\right)$. This channelwise correlation analysis will result in getting features with similar semantic meanings for each modality, as shown in Figure 6, which guides $f_I$ to embed more valuable information related to depth completion.

Using 1-dimensional feature representation would lead to SSS problem in modern deep learning based computer vision task. We introduce the 2-dimensional approach similar to [32] to generate full-rank covariance matrix $\hat{\Sigma}_{sD, sI} \in \mathbb{R}^{m \times n}$, which is calculated as

$$\hat{\Sigma}_{sD, sI} = \frac{1}{C} \sum_{i=0}^{C-1} \left[F_{sD}^i - \mathbf{E}[F_{sD}]\right] \left[F_{sI}^i - \mathbf{E}[F_{sI}]\right]^T, \tag{1}$$

in which we define $\mathbf{E}[F] = \frac{1}{C} \sum_{i=0}^{C-1} F^i$. Besides, we generate covariance matrices $\hat{\Sigma}_{sD}$ (and respective $\hat{\Sigma}_{sI}$) with the regularization constant $r_I$ and identity matrix $\mathbf{I}$ as

$$\hat{\Sigma}_{sD} = \frac{1}{C} \sum_{i=0}^{C-1} \left[F_{sD}^i - \mathbf{E}[F_{sD}]\right] \left[F_{sD}^i - \mathbf{E}[F_{sD}]\right]^T + r_I \mathbf{I}. \tag{2}$$

The correlation between $F_{sD}$ and $F_{sI}$ is calculated as

$$\mathbf{corr}(F_{sD}, F_{sI}) = \left\| (\hat{\Sigma}_{sD}^{-\frac{1}{2}})(\hat{\Sigma}_{sD, sI})(\hat{\Sigma}_{sI}^{-\frac{1}{2}}) \right\|_{\mathbf{tr}}. \tag{3}$$

The higher value of $\mathbf{corr}(F_{sD}, F_{sI})$ represents the higher correlation between two feature blocks. Since $\mathbf{corr}(F_{sD}, F_{sI})$ is an non-negative scalar, we use $-\mathbf{corr}(F_{sD}, F_{sI})$ as the optimization objective to guide training of two feature encoders. To compute the gradient of $\mathbf{corr}(F_{sD}, F_{sI})$ with respect to $\theta_D$ and $\theta_I$, we can compute its gradient with respect to $F_{sD}$ and $F_{sI}$ and then do the back propagation. The detail is showed following. Regarding to the gradient computation, we define $M = (\hat{\Sigma}_{sD}^{-\frac{1}{2}})(\hat{\Sigma}_{sD, sI})(\hat{\Sigma}_{sI}^{-\frac{1}{2}})$ and decompose $M$ as $M = USV^T$ using SVD decomposition. Then we define

$$\frac{\partial \mathbf{corr}(F_{sD}, F_{sI})}{\partial F_{sI}} = \frac{1}{C} \left(2 \nabla_{sDsD} F_{sD} + \nabla_{sDsI} F_{sI}\right), \tag{4}$$

where $\nabla_{sDsI} = \hat{\Sigma}_{sD}^{-\frac{1}{2}} UV^T \hat{\Sigma}_{sRGB}^{-\frac{1}{2}}$ and $\nabla_{sDsD} = -\frac{1}{2} \hat{\Sigma}_{sD}^{-\frac{1}{2}} UDU^T \hat{\Sigma}_{sD}^{-\frac{1}{2}}$. $\frac{\partial \mathbf{corr}(F_{sD}, F_{sI})}{\partial F_{sD}}$ follows the similar calculations as $\frac{\partial \mathbf{corr}(F_{sD}, F_{sI})}{\partial F_{sI}}$ in Equation (4).

## 3.3 Loss Function

We denote our channelwise $2D^2CCA$ loss as $L_{2D^2CCA} = -\mathbf{corr}(F_{sD}, F_{sI})$. We denote the transformed component from sparse RGB to depth domain as $\hat{F}_{sD}$. The transformer loss describes the numerical similarity between RGB and depth domain. We use $L_2$ norm to measure the numerical similarity. Our transformer loss is $L_{trans} = \|F_{sD} - \hat{F}_{sD}\|_2^2$.

We also build another encoder and another transformer network which share weights with the encoder and transformer network for the spare RGB. The input of the encoder is a complementary RGB image. We use features extracted from complementary RGB image to predict features of non-observable depth using transformer network. For the complementary RGB image, we denote the extracted feature and transformed component as $F_{cI}$ and $\hat{F}_{cD}$. Later, we concatenate $F_{sD}$ and $\hat{F}_{cD}$, both of which are 512-channel. We got an 1024-channel bottleneck feature on depth domain. We pass this bottleneck feature into the decoder described in Section 3.1. The output from the decoder is a completed dense depth map $\hat{D}$. To compare the inconsistency between the groundtruth $D_{gt}$ and the completed depth map, we use pixelwise $L_2$ norm. Thus our reconstruction loss is $L_{recon} = \|D_{gt} - \hat{D}\|_2^2$.

Also, since bottleneck features have limited expressiveness, if the sparsity of inputs is severe, e.g. only 0.1% sampled points of the whole resolution, the completed depth maps usually have griding effects. To resolve the griding effects, we introduce the smoothness term as in [36] into our loss function. $L_{smooth} = \|\nabla^2 \hat{D}\|_1$, where $\nabla^2$ denotes the second-order gradients. Our final total loss function with weights becomes

$$L_{total} = L_{2D^2CCA} + w_t L_{trans} + w_r L_{recon} + w_s L_{smooth}. \tag{5}$$

# 4 Experiments

## 4.1 Dataset and Experiment Details

**Implementation details**. We use PyTorch to implement the network. Our encoders are similar to VGG16, without the fully-connected layers. We use ReLU on extracted features after every SAConv operation. Downsampling is applied to both the features and masks in encoders. The transformer network is a 2-layer network, size $3 \times 3$, stride 1, and 512-dimension, with our SAConv. The decoder is also a VGG16-like network using deconvolution to upsample. We use SGD optimizer. We conclude all the hyperparameter tuning in the supplemental material.

**Datasets.** We have done extensive experiments on outdoor scene datasets such as KITTI odometry dataset [12] and Cityscape depth dataset[37], and on indoor scene datasets such as NYUv2 [38] and SLAM RGBD datasets as ICL_NUM [39] and TUM [40].

- **KITTI dataset**. The KITTI dataset contains both RGB and LiDAR measurements, total 22 sequences for autonomous driving use. We use the official split, where 46K images are for training and 46K for testing. We adopt the same settings described in [4, 41] which drops the upper part of the images and resizes the images to $912 \times 228$.

- **Cityscape dataset**. The Cityscape dataset contains RGB and depth maps calculated from stereo matching of outdoor scenes. We use the official training/validation dataset split. The training set contains 23K images from 41 sequences and the testing set contains 3 sequences. We center crop the images to the size of $900 \times 335$ to avoid the upper sky and lower car logo.

- **NYUv2 dataset**. The NYUv2 dataset contains 464 sequences of indoor RGB and depth data using Kinect. We use the official dataset split and follow [4] to sample 50K images as training data. The testing data contains 654 images.

- **SLAM RGBD dataset**. We use the sequences of ICL-NUIM[42] and TUM RGBD SLAM datasets from stereo camera. [40]. The former is synthetic, and the latter was acquired with Kinect. We use the same testing sequences as described in [1].

**Sparsifiers**. A sparsifier describes the strategy of sampling the dense/semi-dense depth maps in the dataset to make them become the sparse depth input for the training and evaluation purposes. We define three sparsifiers to simulate different sparse patterns existing in the real-world applications. Uniform sparsifier uniformly samples the dense depth map, simulating the scanning effect caused by LiDAR which is nearly uniform. Stereo sparsifier only samples the depth measurements on the edge or textured objects in the scene to simulate the sparse patterns generated by stereo matching or direct VSLAM. ORB sparsifier only maintains the depth measurements according to the location of ORB features in the corresponding RGB images. ORB sparsifier simulates the output sparse depth map from feature based VSLAM. We set a sample number for uniform and stereo sparsifiers to control the sparsity. Since the ORB feature number varies in different images, we do not predefine a sample number but take all the depth at the ORB feature positions.

**Error metrics**. We use the error metrics the same as in most previous works. (1) RMSE: root mean square error (2) MAE: mean absolute error (3) $\delta_i$: percentage of predicted pixels where the relative error is within $1.25^i$. Most of related works adopt $i = 1, 2, 3$. RMSE and MAE both measure error in meters in our experiments.

**Ablation studies**. To examine the effectiveness of multi-modal approach, we evaluate the network performance using four types of inputs, i.e. (1) dense RGB images; (2) sparse depth; (3) dense RGB image + sparse depth; (4) complementary RGB image + sparse depth. The evaluation results are demonstrated in Table 1. We could observe that the networks with single-modal input perform worse than those with multi-modal input, which validates our multi-modal design. Besides, we observe that

Table 1: Ablation study of using different data sources on NYUv2 dataset. Dense RGB means we feed in full RGB images. sD means sparse depth generated by 100-point stereo sparsifier . cRGB means complementary RGB image.

| Input data | MAE | RMSE | $\delta_1$ | $\delta_2$ | $\delta_3$ |
|---|---|---|---|---|---|
| Dense RGB | 0.576 | 0.740 | 63.5 | 89.0 | 97.0 |
| sD(100 pts) | 0.524 | 0.700 | 68.1 | 90.2 | 97.0 |
| Dense RGB+sD(100 pts) | 0.479 | 0.638 | 73.0 | 92.4 | 97.7 |
| cRGB+sD(100 pts) | 0.473 | 0.631 | 72.4 | 92.6 | 98.1 |

using dense RGB with sparse depth has similar but worse performance than using complementary RGB with sparse depth. The sparse depth inputs are precise. However, if we extract RGB domain features for the locations where we already have precise depth information, it would cause ambiguity thus the performance is worse than using complementary RGB information. We also conduct ablation studies for different loss combinations in our supplementary material on KITTI and NYUv2 dataset.

Furthermore, we conduct the ablation study with different sparsity on NYUv2 dataset. The stereo sparsifier is used to sample from dense depth maps to generate sparse depth data for training and testing. We show how different sparsity can affect the predicted depth map quality. The results are in Table 2.

Table 2: Ablation study of different sample numbers on NYUv2 using stereo sparsifier.

| Sample# | MAE | RMSE | $\delta_1$ | $\delta_2$ | $\delta_3$ |
|---|---|---|---|---|---|
| 50 | 0.547 | 0.715 | 65.5 | 90.1 | 97.4 |
| 100 | 0.426 | 0.580 | 77.5 | 94.1 | 98.4 |
| 200 | 0.385 | 0.531 | 80.9 | 95.1 | 98.7 |
| 500 | 0.342 | 0.476 | 83.0 | 96.1 | 99.0 |
| 1000 | 0.290 | 0.419 | 87.0 | 97.0 | 99.2 |
| 2000 | 0.242 | 0.352 | 91.3 | 98.2 | 99.6 |
| 5000 | 0.222 | 0.323 | 93.3 | 98.9 | 99.8 |
| 10000 | 0.151 | 0.231 | 96.6 | 99.5 | 99.9 |

## 4.2 Outdoor scene - KITTI odometry and Cityscapes

For KITTI and Cityscapes these two outdoor datasets, we use the uniform sparsifier. For the KITTI dataset, we sample 500 points as sparse depth the same as some previous works. We compare with some state-of-the-art works, [4, 43, 41, 44]. We follow the evaluation settings in these works, randomly choose 3000 images to calculate the numerical results. The results are in Table 3. Next, we conduct experiments using both KITTI and Cityscape datasets. Some monocular depth prediction works use Cityscape dataset for training and KITTI dataset for testing. We choose this setting and use 100 uniformly sampled sparse depth as inputs. The results are shown in Table 4.

## 4.3 Indoor scene - NYUv2 and SLAM RGBD datasets

For NYUv2 indoor scene dataset, we use the stereo sparsifier to sample points. We compare to the state-of-the-art [4] with different sparsity using their publicly released code. The results are shown in Table 5.

Table 3: 500 points sparse depth completion on KITTI dataset.

|  | MAE | RMSE | $\delta_1$ | $\delta_2$ | $\delta_3$ |
|---|---|---|---|---|---|
| Ma *et al.*[4] | - | 3.378 | 93.5 | 97.6 | 98.9 |
| SPN [43] | - | 3.243 | 94.3 | 97.8 | 99.1 |
| CSPN[41] | - | 3.029 | 95.5 | 98.0 | 99.0 |
| CSPN+UNet[41] | - | 2.977 | **95.7** | 98.0 | 99.1 |
| PnP[44] | **1.024** | 2.975 | 94.9 | 98.0 | 99.0 |
| CFCNet w/o smoothness | 1.233 | 2.967 | 94.1 | **98.1** | **99.3** |
| CFCNet w/ smoothness | 1.197 | **2.964** | 94.0 | 98.0 | **99.3** |

Table 4: Depth evaluation results: cap 50m means only taking the depth that smaller than 50m into consideration when doing the evaluation. CS->K means we train the network on the Cityscape dataset and we do the evaluation on the KITTI dataset. Comparing methods all train train/test with 100 pts.

| Methods | Input | Dataset | RMSE | $\delta 1$ | $\delta 2$ | $\delta 3$ |
|---|---|---|---|---|---|---|
| Zhou *et al.* [36] | RGB | CS→K | 7.580 | 57.7 | 84.0 | 93.7 |
| Godard *et al.* [45] | RGB | CS→K | 14.445 | 5.3 | 32.6 | 86.2 |
| Aleotti *et al.* [46] | RGB | CS→K | 14.051 | 6.3 | 39.4 | 87.6 |
| CFCNet(50 pts) | RGB+sD | CS →K | 7.841 | 78.3 | 92.7 | 97.0 |
| CFCNet(100 pts) | RGB+sD | CS→K | **5.827** | **82.6** | **94.7** | **97.9** |
| Zhou *et al.* [36](cap 50m) | RGB | CS→K | 6.148 | 59.0 | 85.2 | 94.5 |
| CFCNet(50 pts, cap 50m) | RGB+sD | CS →K | 6.334 | 79.2 | 93.2 | 97.3 |
| CFCNet(100 pts, cap 50m) | RGB+sD | CS →K | **4.524** | **83.7** | **95.2** | **98.1** |
| CFCNet(50 pts, cap 50m) | RGB+sD | CS→CS | 9.019 | 82.8 | 94.1 | 97.2 |
| CFCNet(100 pts, cap 50m) | RGB+sD | CS→CS | 6.887 | 88.9 | 96.1 | 98.1 |
| CFCNet(100 pts, cap 50m) | RGB+sD | K→K | 3.157 | 91.0 | 97.1 | 98.9 |

Table 5: Comparisons on NYUv2 dataset using stereo sparsifier.

| Sample# | Methods | MAE | RMSE | $\delta_1$ | $\delta_2$ | $\delta_3$ |
|---|---|---|---|---|---|---|
| 100 | [4] | 0.473 | 0.629 | 71.5 | 92.4 | 98.0 |
| 100 | CFCNet | **0.426** | **0.580** | **77.5** | **94.1** | **98.4** |
| 200 | [4] | 0.451 | 0.603 | 73.0 | 93.5 | 98.4 |
| 200 | CFCNet | **0.385** | **0.531** | **80.9** | **95.1** | **98.7** |
| 500 | [4] | 0.384 | 0.529 | 79.2 | 94.9 | 98.6 |
| 500 | CFCNet | **0.342** | **0.476** | **83.0** | **96.1** | **99.0** |

Next, we conduct experiments on SLAM RGBD dataset. We follow the setting in the state-of-the-art, CNN-SLAM [1], and do the cross-dataset evaluation. We train the model on NYUv2 using ORB sparsifier and evaluate on the SLAM RGBD dataset. We use the metric in CNN-SLAM, calculating the percentage of accurate estimations. Accurate estimations mean the error is within ±10% of the groundtruth. The results are in Table 6.

Table 6: Comparison in terms of percentage of correctly estimated depth on two SLAM RGBD datasets, ICL-NUIM and TUM. (TUM/seq1 name: *fr3/long office household*, TUM/seq2 name: *fr3/nostructure texture near withloop*, TUM/seq3 name: *fr3/structure texture far*)

| Sequence# | CFCNet | CNN-SLAM[1] | Laina[47] | Remode[48] |
|---|---|---|---|---|
| ICL/office0 | **41.97** | 19.41 | 17.19 | 4.47 |
| ICL/office1 | **43.86** | 29.15 | 20.83 | 3.13 |
| ICL/office2 | **63.64** | 37.22 | 30.63 | 16.70 |
| ICL/living0 | **51.76** | 12.84 | 15.00 | 4.47 |
| ICL/living1 | **64.34** | 13.03 | 11.44 | 2.42 |
| ICL/living2 | **59.07** | 26.56 | 33.01 | 8.68 |
| TUM/seq1 | **54.70** | 12.47 | 12.98 | 9.54 |
| TUM/seq2 | **66.30** | 24.07 | 15.41 | 12.65 |
| TUM/seq3 | **74.61** | 27.39 | 9.45 | 6.73 |
| Average | **57.81** | 22.46 | 18.44 | 7.64 |

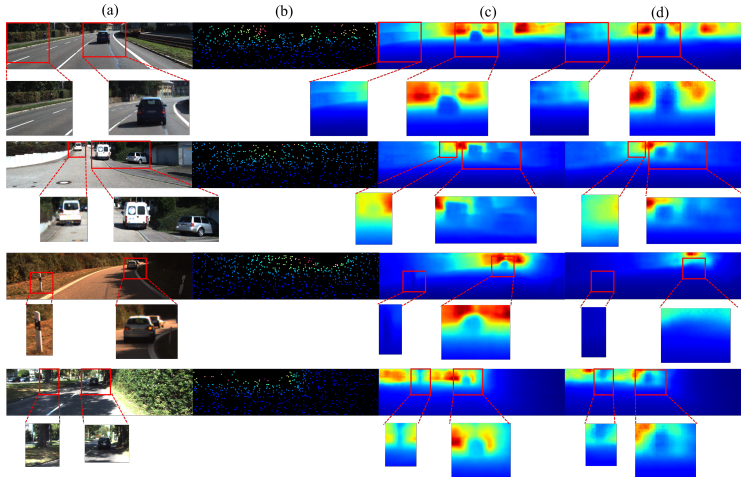

Figure 6: Visual results on KITTI dataset. (a) The RGB image (b) 500 points sparse depth as inputs. (c) Our Completed depth maps. (d) Results from [4].

## 5 Conclusion

In this paper, we directly analyze the relationship between the sparse depth information and their corresponding pixels in RGB images. To better fuse information, we propose 2D$^2$CCA to ensure the most similar semantics are captured from two branches and use the complementary RGB information to complement the missing depth. Extensive experiments using total four outdoor/indoor scene datasets show our results achieve state-of-the-art.

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
