[Supplementary Material · NIPS2019_PaperID2869_Supplementary_Material.pdf]

# Supplementary Material for Submission #2869
# Deep RGB-D Canonical Correlation Analysis For Sparse Depth Completion

## 1 Supplementary Material

### 1.1 Hyper-parameter Configuration

There are five main hyper-parameters in the CFCNet, including three weighting parameters in the Equation (5) in the main paper, learning rate, and batch size. We carefully tune those hyper-parameters for each experiment setting and record the configurations used for evaluation in this subsection.

- **KITTI dataset**. For KITTI dataset, we use 2 different weight settings, $(w_t, w_r, w_s) = (0.5, 0.1, 0.1)$ and $(1, 0.1, 0)$. The batch size is set to be 7, and learning rate is $10^{-3}$. $(w_t, w_r, w_s) = (0.5, 0.1, 0.1)$ is used by the model generating Table 2. $(w_t, w_r, w_s) = (1, 0.1, 0)$ is adopted to generate relative result in Table 3 in the paper.

- **Cityscape dataset**. For Cityscape dataset, We set $(w_t, w_r, w_s) = (1, 0.1, 0.1)$. The batch size is set to be 12, and learning rate is $10^{-3}$.

- **NYUv2 dataset**. For the model evaluated in Table 1, Table 4 and Table 5 in the paper, we set $(w_t, w_r, w_s) = (1, 10, 1)$, batch size to be 4, and learning rate to be $10^{-3}$.

### 1.2 Additional Ablation Studies

We first conduct the ablation study on KITTI dataset, using different loss combinations. We use the official dataset split. We uniformly sample 100 points from the original LiDAR measurements of training dataset as the sparse depth inputs. The whole LiDAR measurements are used as groundtruth data. We show the performance of using different combinations of the proposed loss terms. The results are in Table 1.

Table 1: Ablation study using different loss combinations on KITTI dataset.

| Loss Combination | MAE | RMSE | $\delta_1$ | $\delta_2$ | $\delta_3$ |
|---|---|---|---|---|---|
| $L_{recon}$ | 1.649 | 3.752 | 91.1 | 97.1 | 99.0 |
| $L_{trans} + L_{recon}$ | 1.638 | 3.722 | 91.2 | 97.1 | 99.0 |
| $L_{2D^2CCA} + L_{trans} + L_{recon}$ | 1.612 | 3.676 | 91.2 | 97.1 | 99.0 |
| $L_{2D^2CCA} + L_{trans} + L_{recon} + L_{smooth}$ | 1.615 | 3.733 | 91.2 | 97.2 | 99.0 |

We also conduct the same ablation study on the NYUv2 dataset. We use official dataset split. We used stereo sparsifier to sample 100 points from the dense depth map as our sparse depth inputs. The results are shown in Table 2.

Table 2: Ablation study using different loss combinations on NYUv2 dataset.

| Loss Combination | MAE | RMSE | $\delta_1$ | $\delta_2$ | $\delta_3$ |
|---|---|---|---|---|---|
| $L_{recon}$ | 0.439 | 0.594 | 76.0 | 93.7 | 98.4 |
| $L_{trans} + L_{recon}$ | 0.440 | 0.598 | 76.8 | 93.8 | 98.3 |
| $L_{2D^2CCA} + L_{trans} + L_{recon}$ | 0.428 | 0.581 | 77.6 | 94.2 | 98.4 |
| $L_{2D^2CCA} + L_{trans} + L_{recon} + L_{smooth}$ | 0.426 | 0.580 | 77.5 | 94.1 | 98.4 |

Next, we conduct the ablation study with different sparsity on NYUv2 dataset. We use stereo sparsifier to sample from dense depth maps to create sparse depth data. We show how different sparsity could affect the predicted depth map quality. The results are in Table 3.

Table 3: Ablation study of different sample numbers on NYUv2 using stereo sparsifier.

| Sample# | MAE | RMSE | $\delta_1$ | $\delta_2$ | $\delta_3$ |
|---|---|---|---|---|---|
| 50 | 0.547 | 0.715 | 65.5 | 90.1 | 97.4 |
| 100 | 0.426 | 0.580 | 77.5 | 94.1 | 98.4 |
| 200 | 0.385 | 0.531 | 80.9 | 95.1 | 98.7 |
| 500 | 0.342 | 0.476 | 83.0 | 96.1 | 99.0 |
| 1000 | 0.290 | 0.419 | 87.0 | 97.0 | 99.2 |
| 2000 | 0.242 | 0.352 | 91.3 | 98.2 | 99.6 |
| 5000 | 0.222 | 0.323 | 93.3 | 98.9 | 99.8 |
| 10000 | 0.151 | 0.231 | 96.6 | 99.5 | 99.9 |

## 1.3 Additional Visual Results

We showed several additional visual results. Results of using the stereo sparsifier on NYUv2 dataset are shown in Figure 1. Results of using the ORB sparsifier on NYUv2 dataset are shown in Figure 2. Results of using the ORB sparsifier on ICL-NUIM are shown in Figure 3. We also make a video clip of the depth completion using our CFCNet on KITTI dataset in the other file.

Figure 1: Visual results on NYUv2 dataset using stereo sparsifier. (a) The RGB images (b) 5000 points sparse depth. (c) Results from [1]. (d) Our completed depth maps. (e) Groundtruth dense depth maps.

Figure 2: Visual results on NYUv2 dataset using ORB sparsifier. (a) The RGB images (b) ORB sparse depth. (c) Our completed depth maps. (d) Groundtruth dense depth maps.

Figure 3: Sample results of cross-dataset testing on ICL-NUIM dataset and TUM dataset using ORB sparsifier. Tested model is trained on NYUv2 dataset and is the same model used in Figure 2. (a) The RGB images (b) ORB sparse depth. (c) Our completed depth maps. (d) Groundtruth dense depth maps

## References

[1] Fangchang Ma and Sertac Karaman, "Sparse-to-dense: Depth prediction from sparse depth samples and a single image," in *IEEE International Conference on Robotics and Automation (ICRA)*, 2018.