[Reviews · NeurIPS 2019]

Reviewer 1



Update: After reading all the reviews and the feedback my rating stays the same. Note on visualizations: Please be aware that false colors (especially spectrum LUT) are problematic for people with color blindness. --- Summary: The paper is about depth completion and uses 2D canonical correlation analysis to learn the relationship between color images and depth maps for depth completion. Inputs for this method are a sparse depth map and a color image. Similar to previous methods like "Sparse and Dense Data with CNNs", Jaritz et al. 3DV 2018, the proposed method uses separate encoders for processing depth and color. However, the proposed method separates the color image into a sparse part (masked out where no depth is available) and a complimentary part (masked out where depth is available) and uses a third branch during training. The three encoder branches process the sparse depth map, the complimentary color image and the sparse color image. The last branch, which processes the sparse color image is required for applying the new CCA-based loss during training. The CCA-based loss forces the network to extract features from color images that are highly linearly correlated with features extracted from depth images and vice versa. Besides the CCA loss, the method uses a reconstruction loss on the final depth output and losses to enforce smoothness and numerical range. The architecture uses a binary mask to focus the network on processing the pixels with information, which is inspired by "Sparsity Invariant CNNs", Uhrig et al. 3DV 2017 and "Segmentation-aware Convolutional Networks using Local Attention Masks", Harley et al. ICCV 2017. Finally, the features from the complimentary color branch and depth branch is concatenated and decoded to a dense depth map. Originality: The idea to extend and use deep canonical correlation analysis for a depth completion framework is clearly novel and sets this work apart from previous methods. The network architecture itself is a reasonable combination of existing methods. The most notable changes are a cause of the CCA loss--the main contribution--which is good. The related work section on CCA provides many references and is especially helpful. Quality: Overall the paper does a good job to explain the method and support the claims with experiments. The experiments are diverse and cover multiple different datasets. However, I think some results could be less significant than how they are presented. For instance, the difference between using the complementary RGB image and the dense RGB image is quite small. Adding information about the variance would help the reader to assess the importance of the different input configurations. Further some additional information and evaluation of additional configurations could improve the understanding of the experiments. See Improvements for details. Clarity: There are some small issues with the writing that needs to be fixed but overall the paper reads well. The network for adapting the numerical range is just called "Transformer network", which is ambiguous. Giving a more specific name (maybe "range transform network") would avoid that. There are a few grammar issues. Significance: In my opinion it is likely that future approaches for this task will make use of the CCA-based loss. The good results support the idea presented in this work and prove its significance. I vote for accepting this work. Questions for the authors: Do you backpropagate to the parameters of the sparse depth branch from the 2D CCA loss? Which dataset was used for the results in Table 1? Were the networks trained for the specific input configuration in Table 1?

Reviewer 2



Originality: Somewhat. Using SSA-based loss in combination with a new model architecture is perhaps new but it is yet another combination of existing building blocks. The paper misses an important previous work: [P1] Yinda Zhang, Thomas Funkhouser. Deep Depth Completion of a Single RGB-D Image. CVPR 2018 which should be reviewed and included into the comparison. Quality: The paper execution is reasonable. I would not be able to reproduce it without the source code and data but that is common situation with many DL papers. Clarity: I found the paper quite hard to read, in particular section 3.2 and graphs and tables in Experimental part. Many tables (eg 1,2,3,4,5)have no units. False color visualizations, eg Fig 6, are only very qualitative and do allow to really compare performances of the methods. Significance: Medium. The method seems to work somewhat better than other methods but is not compared with all very relevant methods. The model is a combination of existing elements but reasonably engineered.

Reviewer 3



Summary This paper proposes CFCNet, a method to conduct sparse depth completion from sparse depth and the corresponding RGB, leveraging the relationship between them. The authors first demonstrate the structure of CFCNet. The basic idea is to use convnets to extract features from corresponding RGB and depth, minimizing the channelwise 2D^2CCA loss. A transformer network is introduced to map RGB features to depth features, and it is then used to transform the RGB image into depth. The authors then introduce 2D^2CCA, which is an extension of CCA when the input size is of high-dimensional. Finally, experiments on a bunch of datasets (both indoor and outdoor scenes) show that the proposed method achieves the state-of-the-art performance. Strengths -The depth completion task is important, and the paper provides a method with good performance. -The method is novel. The authors map RGB and depth to two latent spaces where they are highly correlated, so that mapping between the two latent space becomes easier. -The visual results look impressive. As shown in figure 6 and the supplementary, even when testing on real images, results from CFCNet looks impressive, contains much more details compared to previous methods. Minor issues -There are many “directional”s in this paper, e.g., two-directional, and I still do not understand what they mean. -May reorganize all the notations. For example, F_{s_I} is introduced in line 148 but not in figure 2. Putting the notations into figure 2 might help readers to understand section 3.2 easier. Comments after the rebuttal ************************************ Thank the authors for the rebuttal. The visualized results look great and still have space for improvement. I agree with two other reviewers that this is overall a good paper, and my overall score remains the same.

[Author Response · NeurIPS 2019]

We appreciate all the reviewer comments. First, we summarize our contributions. We use a novel $2D^2CCA$ loss to assure similar semantics are captured from the RGB and depth domains; we design a new model structure for sparse depth completion, which exploits the relationship between RGB and depth data; our method achieves the state of the art (SOTA) on several indoor/outdoor datasets.

**Response to common concerns**: First, we will release codes including the data processing tools. Second, we fixed the symbol/annotation inconsistency and added symbols on Figure 2. We will carefully check grammar in the whole paper.

**Response for reviewer #1**:

1. "The difference between using the complementary RGB image and the dense RGB image is quite small." - The difference between known and predicted depth is small (see ablation study, line 229-233 of paper). However, Table 1 validates that using only sparse and complementary information for depth completion is sufficient and can yield better performance than using dense RGB information.

2. "For Table 3 a version of CFCNet with 0 points is missing (only RGB information)" - The core elements of our CFCNet are $2D^2CCA$ and a range transformer. With only RGB information, the network structure would be just an image encoder-decoder, and reduced to a normal FCN. This contradicts our purpose of multi-modal learning. Table 1 with only image information is for the ablation study. It shows that our core elements can improve performance.

3. Need to explain the result reported in the abstract. - The numbers in the abstract are compared with CNN-SLAM. The "13.03" is the number for that sequence. The "58.89" is a typo - the correct number is "64.34" in Table 5. We agree that using the result of a single sequence for comparison could be biased and unfair. Hence, we instead calculate the average performance on SLAM RGBD datasets. We attain a +194% improvement on the datasets.

4. "Which dataset was used for the results in Table 1? Were the networks trained for the specific input configuration in Table 1?" - We used the NYUv2 dataset, as described in the supplementary material. NYUv2 dataset was pre-processed by a 100-point stereo sparsifier and then used as the training data. We will add the information to the body of the paper.

5. "Do you backpropagate to the parameters of the sparse depth branch from the 2D CCA loss?" - Yes, we use Eq. (4) in the paper to backpropagate gradients through both the image branch and sparse depth branch. The whole network is end-to-end trainable.

6. For the K->K experiment the number of sparse depth points is missing. - We state 100 depth points in the title of the table. We will add it to the table body as well.

**Response for reviewer #2**:

We would like to clarify that our proposed loss is CCA-based but not SSA-based.

1. "False color visualizations, eg Fig 6, are only very qualitative and do allow to really compare performances of the methods." - Most of the related works also use false color to visualize results. We follow this practice and use color codes in the same way. Visualizing depth maps with false color is indeed qualitative. Hence we also provide numerical results in paper to compare with others. We will add the numeric data of each example shown in Fig. 6 in the paper.

2. "Many tables (eg 1,2,3,4,5) have no units." - Meters for MAE and RMSE. Percentage for all $\delta$. We will add this info in our final version.

3. "Model seems to work but is just another model." & "The model is a combination of existing elements but reasonably engineered." - We respectfully disagree. Our method is not a simple combination of existing elements. Rather, it is a multi-modal learning framework effectively uses $2D^2CCA$ to exploit the relationship between different types of data. We want to demonstrate to other researchers a meaningful direction, in which the correlation between multi-modal features for depth completion is fully utilized. Since our novel loss is independent from existing elements such as the VGG-16 architecture, future researchers may use our loss along with other models to boost performance.

4. "The paper misses an important previous work: [P1] which should be reviewed and included into the comparison." - We thought it might be inappropriate to compare our results with theirs because the study objectives are different. Our study aims to complete depth when observable measurements are highly limited, while [P1] aims to complete missing data with significantly more observable depth measurements. The authors note that their performance would be negatively impacted if the observed depth samples are fewer than 2000. Our experiments use samples typically fewer than 500.

**Response for reviewer #3**:

1. Meaning of many "directional"s. - "Directional" is synonymous to "dimensional" here, used in some statistical communities. Indeed this may be confusing. We changed "directional" to "dimensional" to avoid confusion.

2. "The authors already demonstrate results when sampling 100 points from LiDAR data, but I am still interested in what will happen when using all LiDAR measurements." - Thank you for your interest. We show numerical results for training and evaluation on KITTI train/val dataset with an additional visual result here (left to right: image, depth completion using full LiDAR points (MAE=0.2822), depth completion using 500 LiDAR points (MAE=0.7871)).

| Sample# | MAE | RMSE | $\delta_1$ | $\delta_2$ | $\delta_3$ |
|---|---|---|---|---|---|
| 500 (from paper,Table 2) | 1.197 | 2.964 | 94.0 | 98.0 | 99.3 |
| full points | 0.596 | 1.568 | 97.5 | 99.3 | 99.8 |



[Meta-Review · NeurIPS 2019]

This is a borderline paper which the reviewers discussed after reading the rebuttal. Overall, the deep CCA method is well-motivated and executed competently; however, the presentation should be improved and some additional comparisons need to be included in the final manuscript.